# Brief Report on Double-Chamber Syringes Patents and Implications for Infusion Therapy Safety and Efficiency

**DOI:** 10.3390/ijerph17218209

**Published:** 2020-11-06

**Authors:** Liliana B. Sousa, Paulo Santos-Costa, Inês A. Marques, Arménio Cruz, Anabela Salgueiro-Oliveira, Pedro Parreira

**Affiliations:** 1Health Sciences Research Unit: Nursing (UICISA:E), Nursing School of Coimbra (ESEnfC), 3046-851 Coimbra, Portugal; paulocosta@esenfc.pt (P.S.-C.); acruz@esenfc.pt (A.C.); anabela@esenfc.pt (A.S.-O.); parreira@esenfc.pt (P.P.); 2Instituto de Ciências da Saúde, Universidade Católica Portuguesa, 4169-005 Porto, Portugal; 3Biophysics Institute, Coimbra Institute for Clinical and Biomedical Research (iCBR) Area of CIMAGO, Faculty of Medicine, CNC.IBILI, Faculty of Medicine, University of Coimbra, Polo das Ciências da Saúde Azinhaga de Santa Comba, 3000-354 Coimbra, Portugal; ines.marques@student.uc.pt

**Keywords:** double-chamber syringes, vascular access devices, flushing, patent review

## Abstract

This review aimed to map the existing patents of double-chamber syringes that can be used for intravenous drug administration and catheter flush. A search was conducted in the Google patents database for records published prior to 28 October 2020, using several search terms related to double-chamber syringes (DCS). Study eligibility and data extraction were performed by two independent reviewers. Of the initial 26,110 patents found, 24 were included in this review. The 24 DCS that were found display two or more independent chambers that allow for the administration of multiple solutions. While some of the DCS have designated one of the chambers as the flushing chamber, most patents only allow for the sequential use of the flushing chamber after intravenous drug administration. Most DCS were developed for drug reconstitution, usually with a freeze-dried drug in one chamber. Some patents were designed for safety purposes, with a parallel post-injection safety sheath chamber for enclosing a sharpened needle tip. None of the DCS found allow for a pre- and post-intravenous drug administration flush. Given the current standards of care in infusion therapy, future devices must allow for the sequential use of the flushing chamber to promote a pre-administration patency assessment and a post-administration device flush.

## 1. Introduction

The administration of different fluids through a Vascular Access Device (VAD) is a common practice in a variety of clinical settings [1,2,3]. Syringes are commonly used in clinical contexts to inject drugs and other fluids into the human body. Typical syringes have a plastic chamber (also known as barrel) with an internal piston that must be moved in order to inject the fluid out through a small opening opposite to the piston. Commonly, syringes are linked to a VAD by a connector (e.g., anti-reflux valve, three-way stopcock) and intravenous drugs are administered into the bloodstream. The most commonly found VAD is the peripheral intravenous catheter (PIVC), which is inserted in patients’ peripheral veins and enables the intravenous administration of fluids, blood products, and drugs directly on the bloodstream [1,2,3].

Intraluminal deposits of medication can form a thrombus or constitute an optimal environment for bacterial growth, both leading to significant complications for patients with a PIVC such as catheter occlusion, phlebitis, or bloodstream infection [3,4,5,6]. Such complications are associated with a higher chance of VAD failure and premature removal, delaying intravenous treatments and significantly increasing care costs [4,5,7].

Likewise, after drug administration, a small amount of medication remains in both the tip of the syringe and in the VAD’s lumen. The recurrence of this can pose several challenges to care efficiency and safety, given that the prescribed amount of medicine will not be entirely administered and can result in the mixing of incompatible drugs if sequential drug administration is performed [8,9,10,11]. 

To avoid such complications, several international guidelines and standards of care on infusion therapy recommend the flushing of VADs before drug administration, allowing for a patency assessment, as well as after every single administration [12,13,14]. To do so, healthcare professionals should perform a pulsatile flushing before and between multiple intravenous drug administrations, followed by the final administration of a cleaning solution while applying positive pressure on the plunger (also referred to as VAD locking) [12,13,14]. This process requires the use of at least two different syringes, one for the preparation and administration of the intravenous drug and another for the flushing solution (0.9% sodium chloride being the most used [3,12,13].

However, the recommended steps for drug administration through a VAD are associated with longer preparation and administration times and the use of larger quantities of medical supplies such as syringes and needles. This may explain why VAD flushing is not always performed by healthcare professionals [15,16,17]. Likewise, the use of several syringes increases the number of catheter manipulations, which can enhance the possibility of catheter-related complications such as phlebitis, dislodgement, or accidental removal. 

Considering this, we can infer that a single syringe capable of accomplish both the drug administration and the flushing procedure would significantly reduce the outlined challenges. In recent years, attention has been given to the potentialities of using double-chamber syringes (DCS) for intravenous drug administration and flushing, which is expected to reduce the number of catheter manipulations, contamination risks, economic costs, and procedural time. DCS involve a complex and costly manufacturing process when compared with single barrel syringes, and only a few such devices have been launched in the medical device’s market [18,19]. However, references to DCS are scattered in the literature, making it difficult for healthcare professionals and managers to access information about the characteristics and potential that these medical devices offer.

Therefore, this study aims to map the existing patents on DCS that can be used for intravenous drug administration according to international guidelines and standards of care, and synthesize their main characteristics.

## 2. Materials and Methods

There is an excess of 60 million patent documents from more than 100 patent issuing authorities around the world [20]. Therefore, conducting a patent review must follow rigorous criteria in order to produce a relevant landscape report. This patent review followed the recommendations of the World Intellectual Property Organization [20] for conducting patent reviews, which included: (i) definition of the countries covered; (ii) definition of the time period covered; (iii) assessment of whether a patent family reduction is viable; and (iv) assessment of whether non-patent literature (NPL) will be included in the analysis. 

Thus, as inclusion criteria, patents had to be published until 1 October 2020, in Portuguese, Spanish, and English, without geographical limitations. As the exclusion criteria, this review did not included patents deriving from the same patent family and non-patent literature, those devices being used for drug reconstitution, not having two chambers that allow for intravenous injection, and not representing DCS. Regarding the search strategy, between March and July 2019 (initial search) and then on the 28 October 2020 (follow-up search), a comprehensive search was conducted in the Google patents database; no preference was giving to conducting individual searches in local patent offices. The search terms used were: “double-chamber syringe”, “double chamber syringe”, “double-chamber system”, “double chamber system”, “dual-chamber syringe”, “dual chamber syringe”, “double-barrel syringe”, “double barrel syringe”, “dual-chamber system”, “dual chamber syringe”, “double-barrel system”, “double barrel syringe”, “two-chamber syringe”, “two-chamber system”, “two-barrel syringe”, “two-barrel system”, “multi-chamber syringe”, “multi-chamber system”, “auto-flush syringe”, “auto flush syringe”, “auto-flush system”, and “auto flush system”. 

Two independent reviewers from the research team screened the patents’ titles, abstracts, and claims prior to retrieving full texts. Each of the eligible papers was assessed independently by two reviewers and data extraction was performed. Any disagreements that arose between the reviewers were resolved through discussion with a third element of the research team. 

## 3. Results

The search identified 26,110 potentially relevant patents. Of these, 16,041 were excluded for being patents of the same device family. The remaining patents were first screened by title and 9956 were excluded. Subsequently, 113 patents were included for abstract and claim analysis by two independent reviewers. Overall, 89 patents were excluded, mainly due being written in another language (*n* = 58), being used for drug reconstitution (*n* = 20), not having two chambers that allow for intravenous injection (*n* = 9), and not representing DCS (*n* = 2). Therefore, 24 patents were included for data extraction and synthesis (Figure 1).

Overall, the included patents were developed between 1975 [21,22] and 2020 [23]. All the included patents have two or more chambers that allow for the administration of multiple fluids (Table 1). Of these, seven patents report the existence of a specific chamber for the flush solution [22,23,24,25,26,27,28], with only one referring that the flushing chamber is pre-filled with 0.9% sodium chloride [24]. Twenty-one devices describe a sequential fluid delivery [22,23,24,25,26,27,28,29,30,31,32,33,34,35,36,37,38,39,40,41,42], in which the solution in the second chamber is only delivered when the first chamber is emptied. One DCSs presents an automatic flush mechanism that is immediately triggered after intravenous drug administration [25]. 

Some of the DCS found (*n* = 7) incorporate a hypodermic needle [21,22,24,26,33,37,42]. In two of these patents, the hypodermic needle is projected inwardly into the first chamber after the first solution is fully dispensed. The needle passes through the barrier to dispense the second solution. Then, the barrier seals the first solution away from the needle to prevent the contamination of the second solution [21,22].

Usually, some of the DCS that we found are structured into a distal and proximal chamber [23,26,31,32,33,34,38,40]. In some DCS, the proximal and distal chambers are separated by a closed valve [26,31,33,34,35,36,37,38] with different opening mechanisms, although the majority is supported on the application of positive differential pressure. Given that some of the found DCSs can be used with pre-filled solutions (which when stored, form gas inside the respective chamber), three of these devices present a gas separator that allows for the safe administration of the intravenous solutions without injecting the formed gas [31,35,38].

Some patents have an inner and an outer chamber for different fluids [27,28,29,30,39,44]. For example, in some patents the outer chamber contains the first fluid/drug and the inner chamber (that moves inside the external chamber) contains the flush solution. Usually, in the locked configuration, the latching mechanism prevents the second piston from longitudinal movement and will only permit the longitudinal motion of the second piston in the unlocked configuration [27,28]. Other DCS patents describe their devices as a syringe that is divided by a vertical plate forming two chambers [21,22,24], designed in the same or different sizes [24]. Three patents [21,22,40] highlight that the double compartment syringe may be filled with two or more medications and pre-packaged in a sterile container by a pharmaceutical company.

Syringes with external chambers (for medication and flushing solution) are referred in some patents to ensure the delivery of both solutions without contamination [42,43,44]. One patent concerns a device that has more than two chambers [43], defined as a multi-chamber injection device that includes multiple syringes arrayed in a circular, linear or other format. This syringe’s chambers may be loaded with different drugs and solutions. Another patent describes an automatic flushing mechanism triggered immediately after the administration of the intravenous drug, ensuring that the amount of drug prescribed is fully administered [44]. Another patent describes the existence of two flow-isolated chambers, where each lumen is isolated to prevent that the solutions in each chamber are contaminated before reaching the syringe’s joint outlet port [42]. Given its design, while using one chamber, the others are mechanically blocked, preventing the accidental mixing of solutions [42]. 

## 4. Discussion

The majority of the DCS patents were designed for drug reconstitution, usually with a freeze-dried drug in a syringe chamber or cartridge, and a reconstitution solution in another chamber. The chambers are usually separated by a middle plunger that allows the diluent to enter the drug chamber for reconstitution. The main advantages of these pre-filled DCS are the increase of dose accuracy, the lower risk of microbial contamination, and the reduction of the procedure time and handling steps required [37,38,39]. Nonetheless, several challenges were identified, such as the freeze-drying and reconstitution processes [39] and the product shelf life [12]. Other DCSs incorporate a parallel post-injection safety sheath chamber for enclosing a sharpened needle tip. These syringes are developed as a protection of the inadvertent contact with the needles after the syringe has been used, which can be extremely important in the reduction of needle-stick injuries and contact with blood-transmitted diseases such as hepatitis or acquired immune deficiency syndrome [40]. Despite the importance of these DCS, they did not comply with the initial purpose of this review, and were thus excluded. 

A closer analysis of the 22 included patents was carried out to understand if they can fulfil the requirements for a secure intravenous drug administration according to current international standards on infusion therapy: initial flushing to ascertain VAD patency, drug delivery, and patency maintenance after/between drug administrations. Although the 22 patents included reported, at least, two independent chambers, only seven had a specific chamber for flushing solution. Several of the included patents identified current challenges in infusion therapy as a significant reason for their development, such as the deliver a 0.9% sodium chloride flush after drug administration to assist pharmacodynamics [26], to ensure that the full dosage been correctly and timely administrated [27,28,29,30]. In fact, the main purpose of VAD flushing involves not only the maintenance of the PIVC patency, reducing build-up of blood or other products on the device’s internal surface, and preventing the mixing of incompatible drugs [45,46,47,48,49], but also ensures that the prescribed amount of drug will be entirely administered [10,50]. Even in the DCS patents that did not specify that one of the chambers was intended for the flush solution, their background sections emphasize the need to flush the vascular access with 0.9% sodium chloride or another physiologically compatible flushing solution [31,35,38]. 

While three of the included DCS patents enable the administration of two different intravenous solutions (one in each independent chamber), they do not meet international requirements on infusion therapy, since catheter flushing can only be performed after drug administration [12,13]. Moreover, none of the included DCS patents enables the assessment of VAD patency before the drug administration, which constitutes a significant gap and unsafe practice. As an example, in the presence of PIVC-related complications such as severe phlebitis, catheter dislodgement, or infiltration/extravasation, the lack of an initial flush to assess catheter patency can result in the administration of potentially irritating or vesicant drugs in the adjacent anatomical tissues and structures, which can lead to the depletion of the peripheral vascular network. The frequent assessment of VAD patency should be considered as an essential practice in infusion therapy, enabling the earlier identification of VAD mal-functioning and prevention of related complications.

Finally, the authors would like to address the limitations of their work. First, the findings of this review are limited to the combination of the database and search terms used, as well as the inclusion criteria defined (e.g., language). Future patent reviews within this topic must address these limitations to conduct a more comprehensive overview of existing innovation. Although patent reviews support decision-making through an data-driven and evidence-based approach [20], this review could not explore which of the found DCS patents were successfully commercialized and implemented in clinical practice. This limitation derives from the lack of studies on DCS development (from concept definition to industrial development and end-user testing) and efficacy in real clinical settings [9].

## 5. Conclusions

This review allowed the mapping of current DCS devices, highlighting their characteristics and potentialities for a safe and efficient intravenous drug administration, considering the importance of performing pre- and post-drug administration flushes. While a few of the reviewed DCS devices enable the administration of a flushing solution during infusion therapy, structural and mechanical barriers prevent health professionals from performing VAD flushing in accordance with the latest guidelines and standards of care in this scope. Future DCS device manufacturers must reflect on these results and develop a syringe that allows health professionals to perform an initial PIVC flush to assess catheter patency, while still being able to complete a final flush after drug administration. Such features will likely increase healthcare professionals’ compliance with current international recommendations while reducing infusion therapy-related costs.

## Figures and Tables

**Figure 1 ijerph-17-08209-f001:**
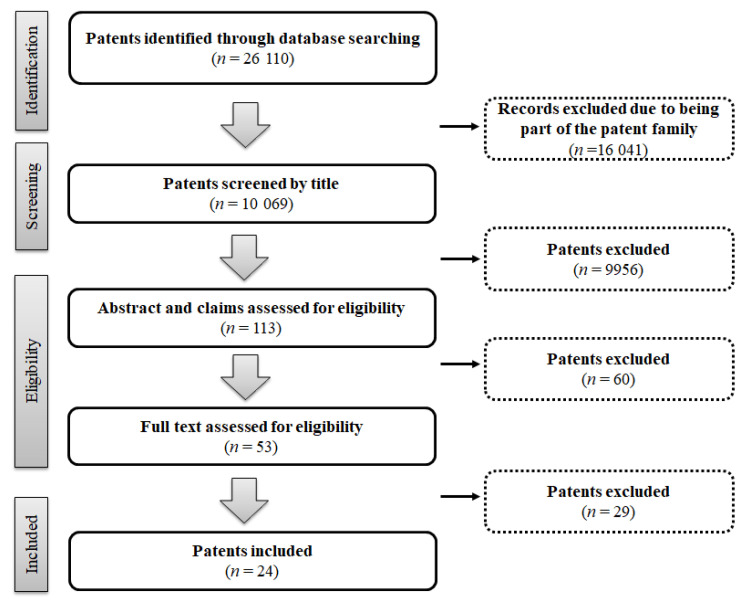
PRISMA flow diagram (adapted).

**Table 1 ijerph-17-08209-t001:** Main characteristics of the double-chamber syringes (DCS) included for review.

Ref.	Patent number	Chambers	Flush Solution	Needle	Sequential Delivery
[21]	US3896805A	2	-	Yes	Yes
[22]	US3923058A	2	-	Yes	Yes
[23]	WO2020077134A1	2	-	-	Yes
[24]	US20070208295A1	2	Yes	Yes	Yes
[25]	US4857056A	2	Yes	-	Yes
[26]	US7077827B2	2	Yes	Yes	Yes
[27]	US8529517B2	2	Yes	-	Yes
[28]	US20090287184A1	2	Yes	-	Yes
[29]	US20120029471A1	2	Yes	-	Yes
[30]	WO2012006555A1	2	Yes	-	Yes
[31]	US6997910B2	2	-	-	Yes
[32]	CA2218734	2	-	-	Yes
[33]	US20020035351A1	2	-	Yes	Yes
[34]	US20180256818A1	2	-	-	Yes
[35]	US9950114B2	2	-	-	Yes
[36]	AU2012202861A1	2	-	-	Yes
[37]	US6723074B1	2	-	Yes	Yes
[38]	US20080319400A1	2	-	-	Yes
[39]	US20100228121A1	2	-	-	Yes
[40]	US20190038836A1	2	Yes	-	Yes
[41]	CA2665697A1	2	-	-	Yes
[42]	US6972005B2	2	-	Yes	-
[43]	US20160030671A1	Multiple	-	-	-
[44]	US6692468B1	2	-	-	-

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
