# Peer review of "Brief Report on Double-Chamber Syringes Patents and Implications for Infusion Therapy Safety and Efficiency"

_ijerph, 2020, doi:10.3390/ijerph17218209_

Round 1

Reviewer 1 Report

Dear authors!

Thank you for letting me review your manuscript which descibes current patents re dual chambered syringes.

The originality of this paper is high and with foresight. It is well written and gives the reader an insight.

I would suggest to add an image or figure of a few DCS models to facilitate to the reader.

Author Response

The authors appreciate the time spent by Reviewer 1 during the assessment of the manuscript. Although we agree that the article would benefit from DCS figures, the IJERPH policy states that authors who wish to include already published figures or images must have the necessary permission from the copyright holder to publish under the CC-BY license.

Thus, to avoid any unwarranted scenarios, and given that contact details in patent files are often outdated and no e-mail address is given by the inventors, we would suggest that no figures are included.

Nonetheless, interested readers can easily search the patent design by searching the patent code. 

Reviewer 2 Report

In this manuscript, the authors summarized the present progress on the DCS patents and analyzed the characteristics of those DCS. As a brief report, the manuscript is acceptable, although the manuscript dismissed some important aspects of the DCS.

Concern 1, the patents represent the future directions of the DCS, namely, they may be commercialized and may not be commercialized. The authors are required to depict the present marketing survey of DCS and incorporate the two parts together.

Concern 2, It seems that the authors also aimed to summarize the targeted clinical implications in the title. However, there is no such content found in the main text. The contents may be integrated into the table and should not be missed.

Concern 3, In figure 1, please make sure the numbers are correct. A lot of numbers are not consistent with the description of the main text. In addition, the formats of the numbers in the main text are versatile and wrong sometimes.

Author Response

The authors would like to thank Reviewer 2 for the time spent during the assessment of this manuscript and the constructive feedback provided. Below are the authors' response to the outlined concerns from Reviewer 2:

  • The patents represent the future directions of the DCS, namely, they may be commercialized and may not be commercialized. The authors are required to depict the present marketing survey of DCS and incorporate the two parts together.

Answer: Although we completely agree with the reviewer's viewpoint, information regarding the patents' commercialization is scarce. In fact, Werk and colleagues (2016) clearly address this challenge, stating "Specific DCS-related literature that deals with these topics is barely available up to current date". Nonetheless, given that we completely agree with Reviewer 2 on this matter, we have addressed this limitation in our discussion section lines 197-202.

  • It seems that the authors also aimed to summarize the targeted clinical implications in the title. However, there is no such content found in the main text. The contents may be integrated into the table and should not be missed.

Answer: We appreciate this comment, but are unsure if we understood this comment correctly since our discussion (lines 167-193) is centred around the shortcomings of current DCS patents when contrasting the existing mechanisms and the clear recommendations for drug administration through a vascular access device. Thus, although the found DCS are innovative, they do not address current challenges in clinical practice concerning the correct flushing and locking of vascular access devices, since no device found allows for a pre- and post-drug delivery catheter flush. 

  • In figure 1, please make sure the numbers are correct. A lot of numbers are not consistent with the description of the main text. In addition, the formats of the numbers in the main text are versatile and wrong sometimes.

Answer: The numbers and Figure were amended, as requested. The authors point out that the new version of the manuscript now includes data from a second follow-up search (as requested by another reviewer), and the number of patents identified in each review phase is now different. 

Reviewer 3 Report

The study entitled “Brief report on double-chamber syringes patents and implications for infusion therapy safety and efficiency”, aimed to map existing patents of double-chamber syringes that can be used for intravenous drug administration and catheter flush. The search was made between March and July of 2019 on Google patents database. It was analysed 22 patents on review. The results found, suggest that the most syringes, were developed for drugs reconstitution, a few patents, were designed for safety purposes, but none, of them, allows a pre- and post-intravenous drug administration flush. To future development of double-chamber syringes it is important that devices allow for the sequential use of the flushing chamber to promote a pre-administration patency assessment and a post-administration device flush.

The topic of this brief report is very interesting and relevant, for the readers of the Int. J. Environ. Res. Public Health, and falls within the journal scope. The importance of this report work, is very relevant for innovation research, but also for the improvement on prevention infections, safety and quality of health cares.

The authors mentioned the methodology aspects required. It is easy-to-read the article. All procedures realized are very clear and well described: participant, concepts, context and methodology phases.

In order to author reflection, I have only one suggestion, that I believe, eventually, would enrich the document. In discussion should have some idea about work limitation/difficulties.

Congratulations for your work.

Author Response

The authors would like to thank Reviewer 3 for the time spent during the assessment of this manuscript and constructive feedback.

Limitations of this review are now addressed in the discussion section, lines 194-202. (Moreover, none of the included DCS patents enables the assessment of VAD patency before the drug administration, which constitutes a significant gap and unsafe practice. As an example, in the presence of PIVC-related complications such as severe phlebitis, catheter dislodgement or infiltration/extravasation, the lack of an initial flush to assess catheter patency can result in the administration of potentially irritating or vesicant drugs in the adjacent anatomical tissues and structures, which can lead to the depletion of the peripheral vascular network. The frequent assessment of VAD patency should be considered as an essential practice in infusion therapy, enabling the earlier identification of VAD mal-functioning and prevention of related complications. )

Reviewer 4 Report

In this systematic review the Google patent database was searched for patents on double-chamber syringes that can be used for IV drug delivery and catheter flushing and that have been published before Jan 1st 2019. Twenty-two patents were included. Most patents allow only for sequential use of flushing after IV drug administration. None of the double-chamber syringes also allows for a catheter flush prior to drug administration to enable catheter patency checking. The authors conclude that it is highly desirable to develop double-chamber syringes that allow for sequential use of the flushing chamber for flushing pre- and post-drug administration. The study is well-performed and the paper is well-written.

One issue with the study is that the search was performed more than a year ago, and only patents published until December 31, 2018 were included. Why did the authors decide to not repeat the database search closer to manuscript submission date?

In addition, several inaccuracies are present in the text that should be addressed.

Abstract line 18: only patents published before Jan 1st 2019 were included

Abstract line 19/20: this is not mentioned in Materials & Methods; please expand in Materials & methods on what is planned to deal with disagreement between the two reviewers.

Results line 99: please check number of excluded patents.

Results line 110: According to table I there are 19 devices with sequential delivery. What is the reason to refer to the literature in two groups, instead of one group of 19 references?

Results lines 142-145: do these sentences refer to [42] instead of [41]?

Discussion line 170: ref. #49 seems inappropriate here, since it refers to a trial proposal and not a primary source of what is claimed in the sentence.

Some references are listed twice (#6 and 7; #22 and 23; #31 and 32). References to # 13, 14 and most patent publications are incomplete (# 22-39; 41-43). In some reference author names are not correct (#24; 40; 41). Is Ref # 35 from 2000 or 2002?  

Author Response

The authors would like to thank Reviewer 4 for the time spent during the assessment of the manuscript and constructive feedback provided. Below are our answers:

  • One issue with the study is that the search was performed more than a year ago, and only patents published until December 31, 2018 were included. Why did the authors decide to not repeat the database search closer to manuscript submission date?

Answer: We completely agree with Reviewer 4. A follow-up search was conducted to include any DCS patents published after December 31st, 2018, using the same methodological approach and search terms. The manuscript now reflects the inclusion of two more DCS patents focused on IV drug preparation/administration. Data and figures were updated accordingly.

  • Abstract line 18: only patents published before Jan 1st 2019 were included.

Answer: Thank you! This was rewritten as suggested.

  • Abstract line 19/20: this is not mentioned in Materials & Methods; please expand in Materials & methods on what is planned to deal with disagreement between the two reviewers.

Answer: The Materials & Methods section now reflects the inclusion of two independent reviewers throughout each phase. 

  • Results line 99: please check number of excluded patents.

Answer: corrected in the text and figure.

  • Results line 110: According to table I there are 19 devices with sequential delivery. What is the reason to refer to the literature in two groups, instead of one group of 19 references?

Answer: This was a mistake on our end, thank you! We have corrected this segment accordingly.

  • Results lines 142-145: do these sentences refer to [42] instead of [41]?

Answer: This line was reviewed and corrected according to each patent description. Thank you.

  • Discussion line 170: ref. #49 seems inappropriate here, since it refers to a trial proposal and not a primary source of what is claimed in the sentence.

Answer: The authors reviewed the initial reference and agree with Reviewer 4. The reference was removed, thank you!

  • Some references are listed twice (#6 and 7; #22 and 23; #31 and 32). References to # 13, 14 and most patent publications are incomplete (# 22-39; 41-43). In some reference author names are not correct (#24; 40; 41). Is Ref # 35 from 2000 or 2002?  

Answer: We reviewed the highlighted references and made amendments. The authors thank Reviewer 4 for carefully reviewing this section.

Round 2

Reviewer 4 Report

The authors have addressed all my comments adequately and satisfactorily.